# Innate-like self-reactive B cells infiltrate human renal allografts during transplant rejection

Yuta Asano [1], Joe Daccache [2], Dharmendra Jain[3], Kichul Ko[1], Andrew Kinloch[1], Margaret Veselits[1], Donald Wolfgeher[4], Anthony Chang [5], Michelle Josephson [6], Patrick Cunningham[6], Anat Tambur[7], Aly A. Khan[5], Shiv Pillai [8], Anita S. Chong[3] & Marcus R. Clark [1 ✉]

Intrarenal B cells in human renal allografts indicate transplant recipients with a poor prognosis, but how these cells contribute to rejection is unclear. Here we show using single-cell RNA sequencing that intrarenal class-switched B cells have an innate cell transcriptional state resembling mouse peritoneal B1 or B-innate (Bin) cells. Antibodies generated by Bin cells do not bind donor-specific antigens nor are they enriched for reactivity to ubiquitously expressed self-antigens. Rather, Bin cells frequently express antibodies reactive with either renal-specific or inflammation-associated antigens. Furthermore, local antigens can drive Bin cell proliferation and differentiation into plasma cells expressing self-reactive antibodies. These data show a mechanism of human inflammation in which a breach in organ-restricted tolerance by infiltrating innate-like B cells drives local tissue destruction.

[1] Section of Rheumatology and The Knapp Center for Lupus and Immunology Research, Department of Medicine, The University of Chicago, Chicago, IL, USA. [2] Transplantation Research Center, Renal Division, Brigham and Women's Hospital, Harvard Medical School, Boston, MA, USA. [3] Section of Transplant, Department of Surgery, The University of Chicago, Chicago, IL, USA. [4] Department of Molecular Genetics and Cell Biology, The University of Chicago, Chicago, IL, USA. [5] Department of Pathology, The University of Chicago, Chicago, IL, USA. [6] Section of Nephrology, Department of Medicine, The University of Chicago, Chicago, IL, USA. [7] Transplant Immunology Laboratory, Northwestern University, Chicago, IL, USA. [8] Ragon Institute of Massachusetts General Hospital, Massachusetts Institute of Technology, and Department of Medicine, Harvard Medical School, Boston, MA, USA. ✉email: mclark@uchicago.edu

In germinal centers (GCs), spatial and molecular orchestration of clonal expansion, somatic hypermutation (SHM), and selection drive production of high-affinity antibodies and immunological memory[1,2]. In many chronic inflammatory and autoimmune diseases, GC-like structures form in afflicted organs (tertiary lymphoid structures, TLSs)[3,4]. TLSs are often associated with hallmarks of antigen-driven B cell selection including local clonal expansion and SHM. However, in most cases the antigens driving in situ B cell selection in TLSs are not known. Furthermore, early in the course of inflammation infiltrating T and B cells are not usually organized into histologically obvious TLSs. Indeed, diffuse lymphocytic infiltrates and T:B aggregates are more common than TLSs in most diseases[5–9]. It remains unclear if local antigens shape in situ lymphocyte repertoire in these disease states. Therefore, our fundamental understanding of in situ adaptive immunity, in both acute and chronic inflammation, is incomplete.

An example of inflammation and progressive organ dysfunction is provided by renal allograft rejection. Acute and early chronic rejection is associated with disorganized lymphocytic infiltrates or T:B cell aggregates while progression to end-stage rejection can be associated with TLSs[10,11]. B cell infiltrates appear important in acute rejection as they predict poor graft survival[12–15]. Furthermore, in mice and humans, B cell depletion mitigates rejection[16,17]. These observations suggest an important role for in situ adaptive immunity, and infiltrating B cells, in allograft rejection.

One clear pathogenic function of B cells is the secretion of donor-specific antibodies (DSAs) that recognize donor human leukocyte antigen (HLA). Serum DSAs strongly predict early onset of allograft rejection[18–20]. The source of these DSAs is not known. A study of a single infected end-stage kidney explant suggested infiltrating B cell infrequently expressed DSAs[21]. It is not known if infiltrating B cells express DSAs in ongoing rejection.

In both mice and humans, renal transplant rejection can be associated with loss of tolerance and serum antibodies to self-antigens[22]. However, it is not known how and where tolerance to self is broken in allograft recipients who do not have an underlying autoimmune disease. In mice, activation of innate immune pathways is sufficient to break B cell tolerance[23–26]. However, it is not clear how this paradigm applies to humans. Therefore, in ongoing renal allograft rejection, the antigens driving in situ B cell selection, and the magnitude of that selection, remain unknown.

Herein, using single-cell RNA sequencing (scRNA-seq) we report that in allograft rejection, intrarenal B cells have a unique transcriptional state that resembles mouse B1 innate-like (Bin) cells. Bin cells are not a source of DSAs. Bin cells generate renal or inflammation-specific antibodies and can give rise to plasma cells selected by local antigens. These results demonstrate how intrarenal B cells drive local inflammation in allograft rejection and provide an example of how inflammation can give rise to organ-restricted autoimmunity.

## Results

### Distinct transcriptional states in activated intrarenal and tonsil B cells.
We first sorted CD45$^+$ DAPI$^-$ Calcein$^+$ CD19$^+$ CD38$^+$ activated B cells from five renal allograft biopsies and four tonsillectomy samples (Fig. 1a and Supplementary Fig. 1a). A paired biopsy from each renal allograft patient was reviewed by a blinded renal pathologist. The presence of B cells and C4d deposition was examined by immunohistochemistry and serum assayed for DSAs. All biopsies displayed diffuse infiltrates and/or lymphocyte aggregates without TLS. Furthermore, all five biopsies had features of either chronic or chronic and active antibody-mediated

rejection (AMR). Pathological features, including Banff scores[27], clinical characteristics and follow-up for each patient are provided in Supplementary Table 1.

Sorted B cells were then subjected to scRNA-seq (Smart-Seq2)[28]. We excluded cells which had less than 3,000 or more than 15,000 expressed genes (Supplementary Fig. 1b). We removed cells which had low expression of immunoglobulin (Ig) constant region genes. After quality control (QC), 655 renal and 129 tonsil B cells were used for subsequent analyses. Batch effects from separate sequencing runs were normalized using External RNA Control Consortium (ERCC) spike-in control and RUVSeq R package[29] (Supplementary Fig. 1c, d).

We first mapped sequenced cells onto a t-distributed stochastic neighbor embedding (t-SNE) space (Fig. 1b). Renal B cells formed one diffuse cluster while tonsil B cells formed two distinct clusters, one of which overlapped with the kidney cluster and the other that was distinct. This clustering was not due to batch-associated differences, suggesting that B cells in renal allograft and tonsil had distinct transcriptional profiles. Moreover, B cells from all five renal biopsies were similarly distributed in the t-SNE space suggesting that renal allograft-infiltrating B cells had a similar transcriptional profile across patients regardless of their histological or clinical features.

Intrarenal B cells could be separated based on Ig class switching (Fig. 2a). Here, B cells expressing IgM or IgD as the most highly expressed Ig isotype were categorized as "unswitched", and those expressing either IgG, IgA, or IgE as "switched." Unswitched cells composed about 70% of both renal and tonsil B cells, and the remaining class-switched cells mostly expressed IgG or IgA (Supplementary Table 2). Regardless of Ig class, switched cells from each tissue source were similarly distributed in the t-SNE space (Fig. 2b, c). The two tonsil B cell clusters were also distinguished by their Ig class. Unswitched tonsil B cells largely overlapped with unswitched renal B cells whereas switched tonsil cells formed a distinct cluster. The differences between clusters were also apparent when examining Pearson correlation coefficients of high-variant genes (Supplementary Fig. 2a). These clustering differences persisted when Ig constant region genes were removed (Supplementary Fig. 2 b). These data suggest that B cells in rejected renal allograft and tonsil tissue are similar prior to class-switch recombination but diverge thereafter.

Comparison across tissue sources and Ig class-switch states identified 2,855 differentially expressed genes (DEGs) which could be divided into six hierarchical clusters (Fig. 2d and Supplementary Data 1). Cluster 1 included genes enriched in unswitched tonsil B cells, clusters 2 and 3 genes enriched in intrarenal cells, cluster 4 genes enriched in intrarenal and tonsil switched cells, cluster 5 genes enriched in tonsil switched cells and cluster 6 genes enriched in tonsil B cells. A pathway enrichment analysis based on Gene Ontology (GO) and Kyoto Encyclopedia of Genes and Genomes (KEGG) databases revealed specific biological pathways were enriched in most clusters (Fig. 2e).

Many of the GO and KEGG pathways enriched in cluster 2 were related to innate receptors and signaling pathways including the pattern recognition receptors NLRP1, NOD1, TLR2, and TLR7 (Supplementary Table 3). Therefore, we next examined if, globally, clusters 2 and 3 were enriched in GO genes termed "innate immune response". When we calculated a sum of scaled expression values for these genes, intrarenal B cells, especially those that were class-switched, had higher values than tonsil (Supplementary Fig. 2c). This enrichment of innate immune response genes was consistent across all patients (Supplementary Fig. 2d). These data reveal an enrichment for innate immune response genes in intrarenal B cells.

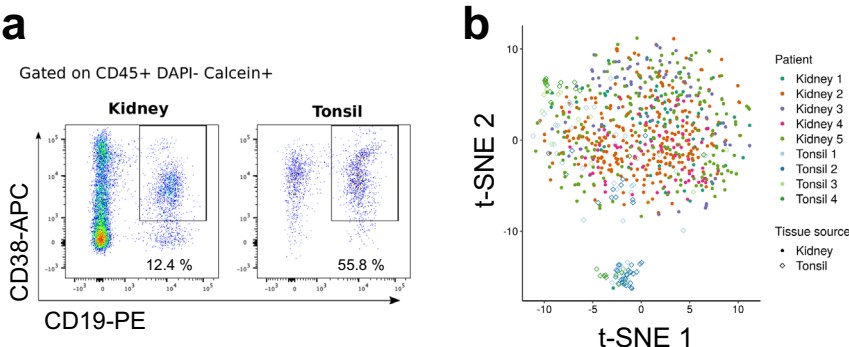

**Fig. 1 Sorting and scRNA-seq of activated B cells in renal allograft and tonsil. a** Gating scheme for single-cell sorting of CD19+ CD38+ activated B cells in renal allograft and tonsil samples. **b** A t-SNE plot of scRNA-seq. Color and shape respectively indicate patients and tissue sources from which cells were derived.

Clusters 2 and 3 were enriched in interferon (IFN)-related pathways including *NLRC5, IFNAR2, IRF1,* and *STAT*[30]. These clusters also contained several cytokine ligands and receptors: *IL15, TNFRSF1B,* and *TNFRSF13B* (Fig. 2f). *TNFR13B* encodes TACI, a receptor for BAFF overexpression of which is associated with renal allograft rejection[31,32].

Consistent with a previous report, the anti-apoptotic factor *BCL2* was enriched in cluster 2 (Fig. 2g)[33]. Many of the pathways enriched in cluster 2, including *BCL2*, TLRs, interferons and cytokines are directly repressed by BCL6[34]. Indeed, the expression of *BCL6* was lower in renal B cells (Fig. 2h), as well as another transcriptional repressor *BACH2*, which shares targets with BCL6[35] (Fig. 2i).

*BCL6* and *BACH2* were preferentially expressed in class-switched tonsil B cells. These cells were enriched in several pathways that have previously been ascribed to GC B cells including proliferation and somatic hypermutation. Notably, *AICDA* was expressed in class-switched tonsil B cells but not significantly in other B cell populations (Fig. 2j). These results indicate that intrarenal class-switched B cells lack the essential transcriptional features of GC B cells.

Neither gene cluster 3 nor 4 demonstrated upregulation of specific GO pathways. However, examination of individual differentially expressed genes revealed potentially important differences. Most notable was *AHNAK* (Fig. 3a). *AHNAK* mRNA levels were far higher in intrarenal B cells compared to tonsil regardless of Ig class switch (Fig. 3b). This corresponded to detectable expression of the AHNAK protein in intrarenal but not tonsil B cells (Fig. 3c). Interestingly, within mouse B cell subsets, *Ahnak* is preferentially expressed in peritoneal cavity B1a and B1b cells (Immgen, Fig. 3d)[36]. This expression pattern is shared with murine homologues of several other cluster 3 genes, such as *ITGAM* and *VIM* (Supplementary Fig. 2e, f). Therefore, we examined whether cluster 3 was enriched for genes having an *AHNAK* covariant expression pattern.

We identified 333 mouse genes whose expression pattern in peripheral B cell populations was similar to *Ahnak* (correlation coefficient ≥ 0.8) (Fig. 3d and Supplementary Data 2). The *Ahnak*-covariant murine genes corresponded to 293 human homologues. These human genes were enriched for cell adhesion and lymphocyte activation pathways, as well as an innate immune pathway related to lipopolysaccharide responses (Fig. 3e). *AHNAK*-covariant genes were highly enriched in cluster 3 and to a lesser degree in clusters 2 and 4 (Fig. 3f). Furthermore, these differences were not solely dependent upon *AHNAK* as they persisted when *AHNAK* was removed (Supplementary Fig. 2g). These results suggest that *AHNAK*-covariant genes are a signature of intrarenal B cell activation.

Although the *AHNAK*-covariant genes were enriched in cluster 3, they represented only 5% of the cluster. Therefore, we next tested if cluster 3 was generally enriched for peritoneal cavity B1 cell-associated genes. First, we converted 2,855 differentially expressed human genes to their murine orthologs. Then, in the Immgen data, we compared murine B cell subsets for the expression of genes in each human gene cluster. For each gene cluster, we calculated a sum of gene expression values scaled across the B cell subsets. This analysis demonstrated that genes in cluster 3 were preferentially expressed in murine peritoneal cavity B1 cells (Fig. 3g, Supplementary Table 3). These results suggest that cluster 3, containing genes highly expressed in the graft-infiltrating B cells, has a gene signature of peritoneal B1 cells. We refer to these unique human innate B cells as Bin cells.

Double negative (CD27-IgD-, DN) B cells have been identified in tissue inflammation including lupus nephritis[37]. They are characterized by increased expression of *TBX21* (T-bet), *ITGAX* (CD11c), *TLR2,* and *TLR7*. Although both *TLR2* and *TLR7* were preferentially expressed by intra-graft B cells, we found no difference between intrarenal and tonsil B cells in the overall expression of 26 genes associated with DN cells (Supplementary Fig. 3a)[37,38].

We also examined the HIF-1 pathway, which is upregulated in situ in both human and murine lupus nephritis[39,40]. However, HIF-family genes were not upregulated in intrarenal B cells compared with tonsil B cells (Supplementary Fig. 3b–f). Overall, these data demonstrate that class-switched intrarenal B cells in allograft rejection have a unique transcriptional profile reminiscent of innate B1 cells.

Consistent with this innate-like transcriptional phenotype, class-switched intrarenal B cells also preferentially expressed the innate cytokine IL-15 (Fig. 4a). To assess if differential mRNA expression was reflected in the protein abundance, we stained tonsil and renal allograft rejection tissue with anti-IL15 antibodies as well as antibodies specific for the IL-15 receptor chain, IL-15RA. There was no detectable IL-15 expression in tonsil germinal centers whereas expression was readily detected in infiltrating B and other immune cells in rejected renal allografts (Fig. 4b). IL-15RA was moderately expressed in both tonsil and renal graft tissue. However, it was more abundant in renal tubular cells than infiltrating immune cells, suggesting that IL-15 secreted by B cells might be captured by tubular cells for presentation to immune cells[41].

Compared to Ig switched tonsil cells, intrarenal B cells differentially expressed several migration and adhesion-related genes including *CD24, CD44, CD55, ITGA4, COL4A4, CCR6, VIM, SIPR1,* and *SIPR2* (Fig. 4c). The differential expression of these molecules suggests that intrarenal B cells respond to

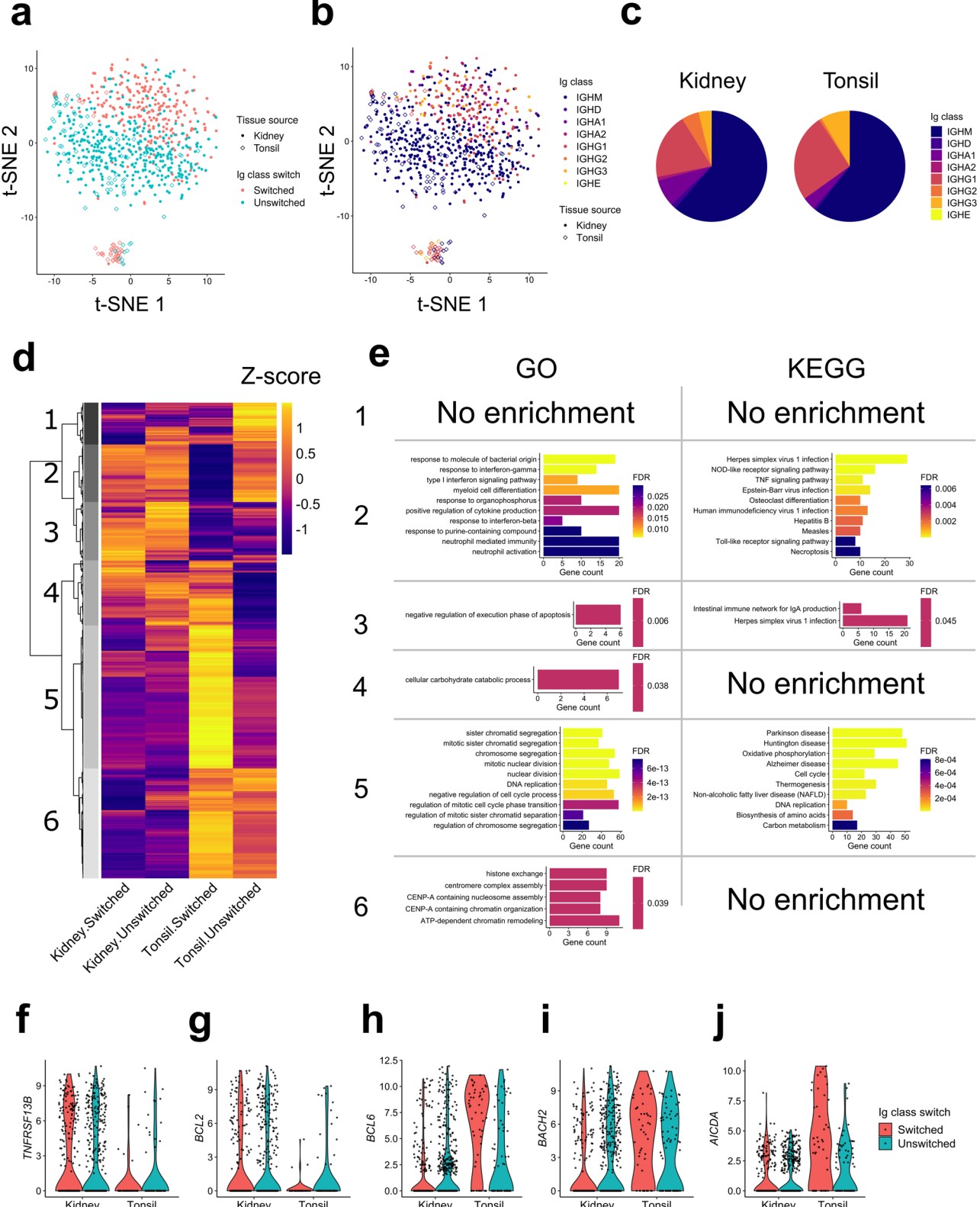

**Fig. 2 Transcriptional state of class-switched intrarenal B cells. a**, **b** t-SNE plots as in Fig. 1b. Color indicates Ig class-switch state (**a**) or expressed Ig classes (**b**). The cells were categorized as "switched" if their most highly expressed Ig heavy-chain genes were either IgG, A or E, and categorized as "unswitched" otherwise. **c** Pie charts showing the distribution of Ig class in intrarenal and tonsil B cells. **d** A heatmap showing hierarchical clustering of the 2,855 DEGs. Mean expression values were calculated for each four cell populations based on their tissue source and Ig class-switch state and then converted to Z-scores. **e** Enrichment of GO terms and KEGG pathways in the five gene clusters. At most 10 most significantly enriched pathways were shown per cluster. **f–j** Violin plots showing RNA expression of *TNFRSF13B* (f), *BCL2* (g), *BCL6* (h), *BACH2* (i), and *AICDA* (j).

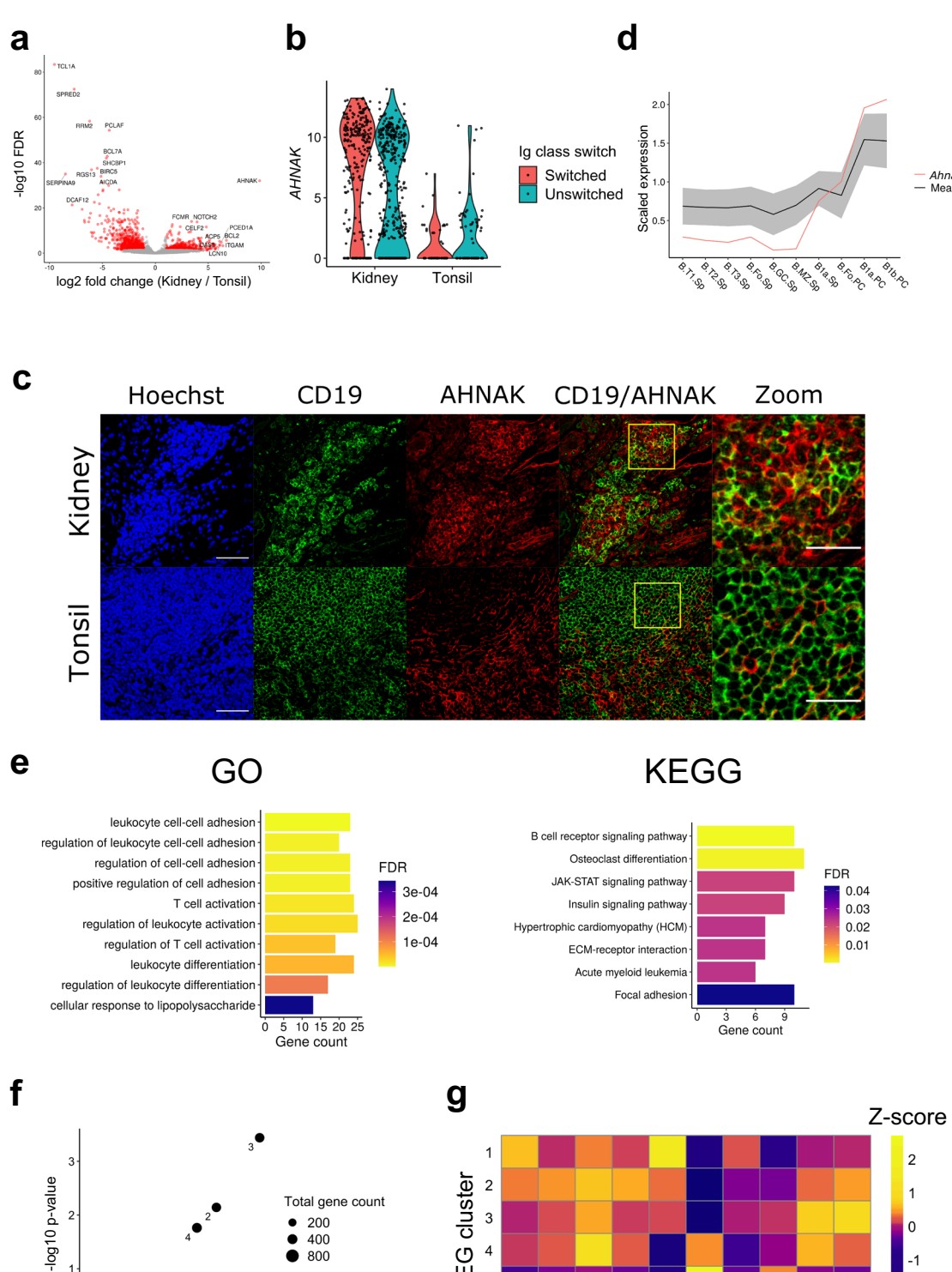

different localization signals than GC B cells. For example, S1PR1 controls B cell egress from lymphoid organs while S1PR2 coordinates GC B cell migration[42,43]. The relatively high expression of *S1PR1* and low expression of *S1PR2*, which is the inverse of what is observed in GCs, is consistent with fundamentally different mechanisms of retention in inflamed renal tissue.

**Serum DSAs are not predictive of intrarenal B cell phenotype**. In renal allograft rejection, the presence of serum DSAs predicts a worse clinical outcome[18–20]. We examined if the presence of serum DSAs was reflected in differences in the transcriptional programs of intrarenal B cells. Since our initial cohort had only one DSA-positive patient, we obtained three additional renal biopsies (two from DSA-positive patients and one from a DSA-negative

**Fig. 3 Intrarenal B cells have an innate-like gene signature. a** A volcano plot showing DEGs between Ig class-switched intrarenal and tonsil B cells. Genes expressed higher in intrarenal B cells are shown on the right side of the plot. **b** A violin plot demonstrating RNA expression of *AHNAK*. **c** Staining images of AHNAK with nuclei (Hoechst) and a B-cell marker CD19 in rejected renal allograft and tonsil. The high-magnification panel corresponds to the yellow square on the merged panel. Scale bars indicate 50 μm or 25 μm (high-magnification panel). Staining has been tested on tissues from two patients and a representative result is shown. **d** Expression of 333 murine genes that correlate with *Ahnak* in Immgen. The mean value of the 333 *Ahnak*-covariant genes (including *Ahnak* itself) is shown as the black line with the gray shade indicating standard deviation. Expression of *Ahnak* is the red line. T: transitional, Fo: follicular, GC: germinal center, MZ: marginal zone, Sp: spleen, and PC: peritoneal cavity. **e** Enrichment of GO terms and KEGG pathways in the 293 AHNAK-covariant genes. At most 10 most significantly enriched pathways are shown. **f** Enrichment of the AHNAK-covariant genes in each gene cluster from Fig. 2d. **g** A heatmap showing DGE scores, a sum of scaled expression levels of each gene cluster within each murine B cell subset in Immgen data. Each row and column represents the gene clusters found in Fig. 2d and the murine B cell subpopulations. DEG scores were scaled by row to obtain Z-scores.

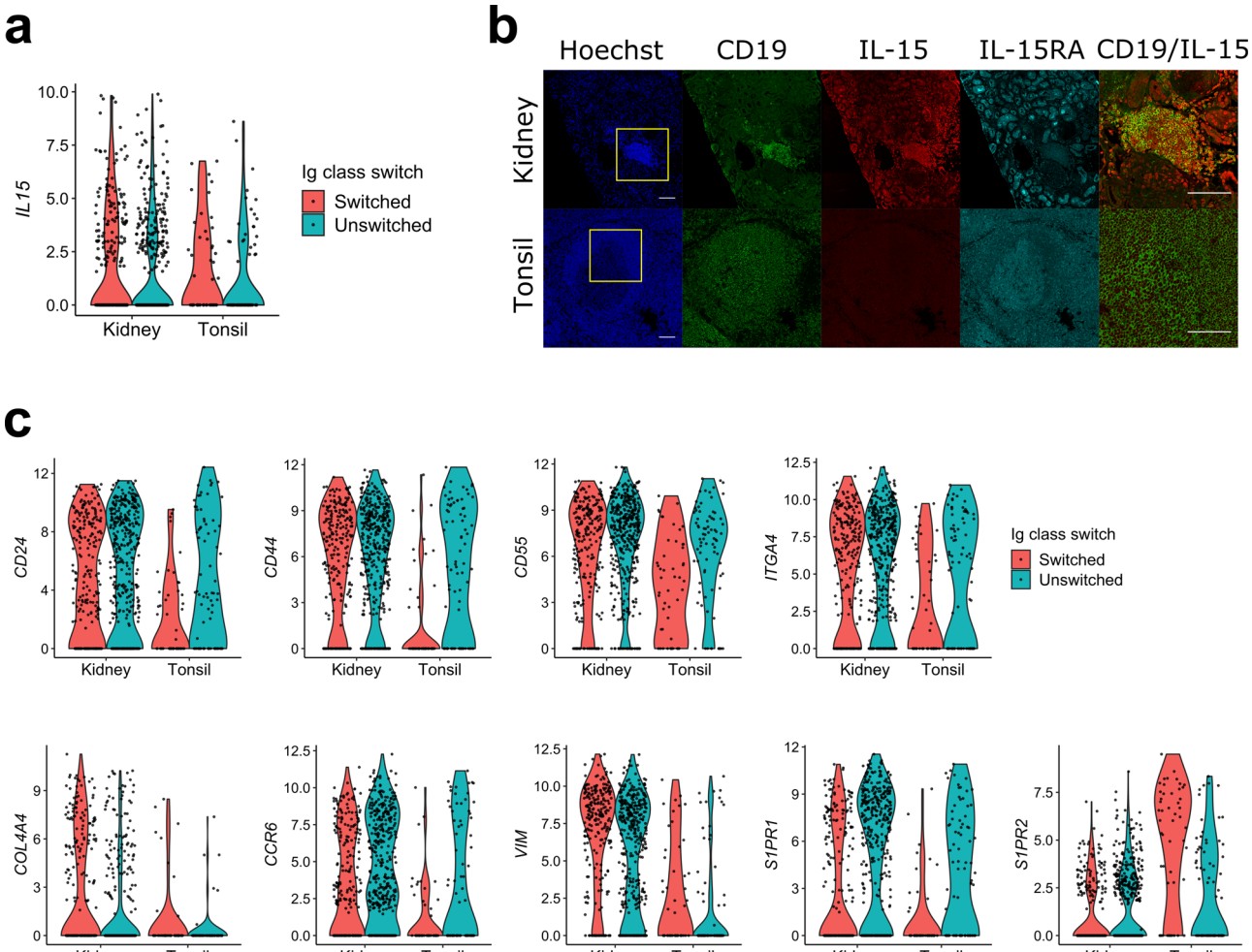

**Fig. 4 Differential expression of innate and localization receptors. a** A violin plot showing the expression level of *IL15*. Cells were grouped by their tissue source and color indicates Ig class-switch state. **b** Staining images of nuclei (Hoechst), CD19, IL-15, and IL-15RA in rejected renal allograft and tonsil tissues. The merged CD19/IL-15 panels were a magnification of the yellow box in merge panels. Scale bars indicate 100 μm. Staining has been tested on tissues from two patients and a representative result is shown. **c** Violin plots of indicated genes that were differentially expressed in intrarenal B cells compared to class-switched tonsil B cells.

patient) as well as three additional tonsillectomy samples. From these samples, activated B cells were sorted and subjected to scRNA-seq (Supplementary Table 1).

Across the integrated dataset, seven of eight biopsies demonstrated mild or moderate interstitial fibrosis and tubular atrophy (Supplementary Table 1). All had chronic and/or active AMR. Only one patient rapidly progressed to renal failure within a month of biopsy. Therefore, the biopsies were comparable and largely reflected rejection in functioning, and not end-stage, renal grafts.

After filtering for both gene coverage and Ig expression, we obtained an additional 513 cells to integrate with those of the first cohort (Supplementary Fig. 4a). The second cohort had a lower sequence coverage compared to the first cohort (Supplementary Fig. 4b). Since normalizing scRNA-seq data with different sequencing depths by a single-scaling factor (e.g. total mapped reads) could introduce bias, we first normalized our data by SCTransform implemented in Seurat[44–46]. The data were then integrated by ComBat[47] to negate batch effects between cohorts (Supplementary Fig. 4c, d).

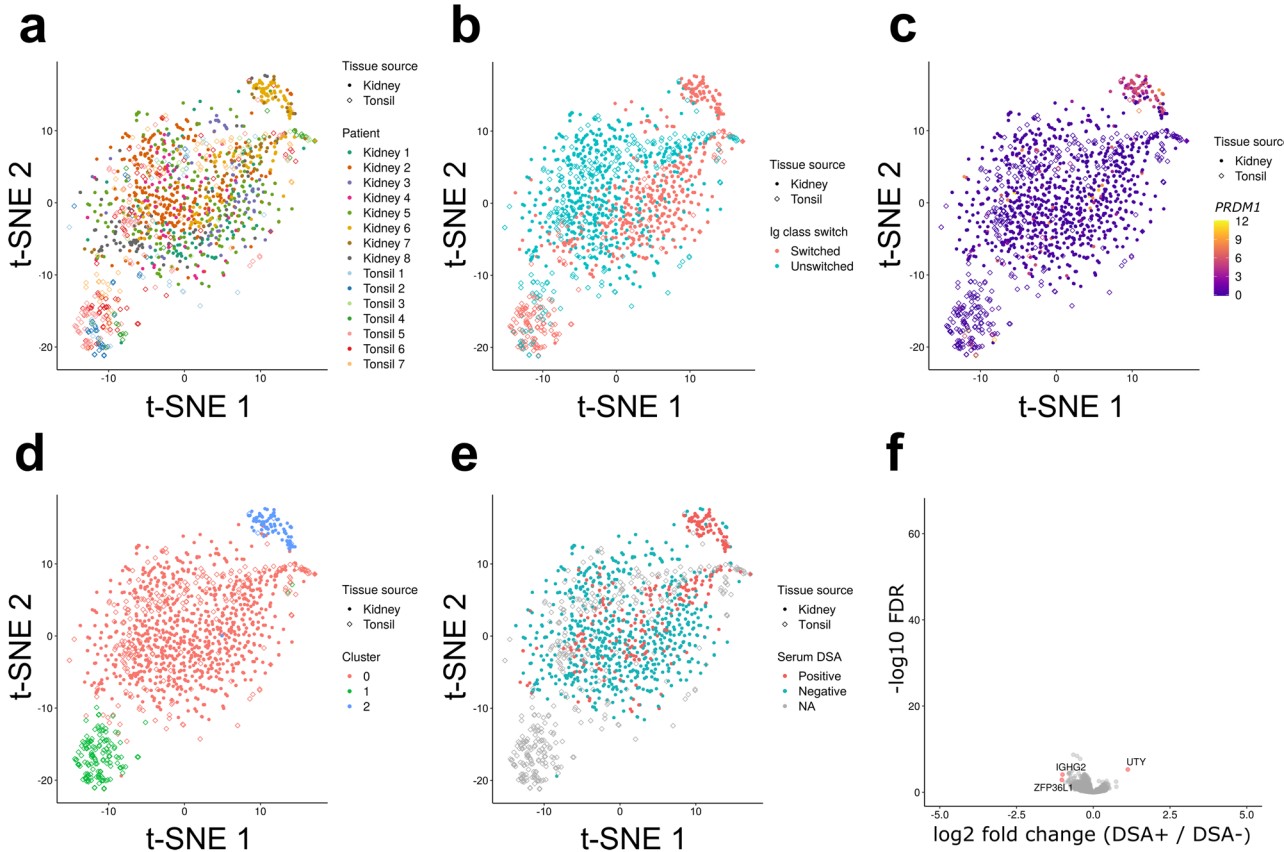

**Fig. 5 B cells are similar between serum DSA-positive and negative patients. a–e** t-SNE plots of the integrated data of the two cohorts. Shape indicates tissue source, and color indicates patients (**a**), Ig class-switch state (**b**), *PRDM1* expression (**c**), clusters assigned by Seurat (**d**), and serum DSA positivity (**e**). **f** Volcano plots showing differential gene expression in non-plasma B cells, comparing DSA-positive and negative patients. Shown on the right side of the plots are genes upregulated in B cells from DSA-positive patients. Genes above the significance threshold were colored in red, and names of top-hit genes were labeled.

The resulting integrated data had a similar t-SNE projection to that of the first dataset (Fig. 5a). As in the previous analysis, class switched B cells were separate whereas unswitched cells clustered together (Fig. 5b). Pearson correlation coefficients for high-variant genes confirmed the relative population differences observed in t-SNE plots (Supplementary Fig. 4e). Also, Pearson correlation analysis between the integrated cohort and the first cohort of five patients revealed strong similarities between comparable populations (Supplementary Fig. 4f). These data reveal clear and consistent differences between B cell populations across cohorts.

In addition to the cell populations observed in the first patient cohort, there was a new discrete cluster which highly expressed *PRDM1* indicative of plasma cells (Fig. 5c, d). This population was mostly derived from kidney patient 6 and, to a lesser degree, patient 7. A few plasma cells were also detected in biopsies from patients 2, 5, and 8. Notably, plasma cells were observed in biopsies from two DSA-positive and three DSA-negative patients. Therefore, intrarenal plasma cells were not a unique feature of DSA-positive patients.

The other B cell populations did not cluster depending on patients' serum DSA positivity (Fig. 5e). Comparing gene expression of the non-plasma cells from the DSA-positive and DSA-negative patients, we could identify only three genes differentially expressed (Fig. 5f). These results suggest that there are no substantial transcriptional differences in graft-infiltrating B cells from DSA-positive and DSA-negative patients.

**In situ immunoglobulin repertoire in renal allograft rejection.** A central question is whether intra-graft B cells are selected for alloreactivity. Therefore, we first used nested polymerase chain reaction (PCR) to amplify immunoglobulin gene variable regions from the cDNA of single B cells isolated from seven kidney biopsies and two tonsil samples[48]. Obtained sequences were aligned to IMGT reference using IMGT/HighV-QUEST and analyzed for somatic mutations. In total, we identified full-length Ig heavy chain variable regions from 457 B cells in seven rejection patients and 77 in two tonsillectomy patients (Table 1)[49,50]. Overall, immunoglobulin mutation burden was similar between rejection and tonsillectomy samples (Fig. 6a). This was true for both B cells expressing unswitched and switched immunoglobulin genes. In general, unswitched B cells had less of a mutation burden than switched cells. However, in intrarenal B cells there were a small fraction of both unswitched and switched cells that had very high (>60) frequencies of mutations.

Next, we assessed clonal relationships among the sequenced antibodies. We tested whether the B cells were clonally related, defined as sharing the same variable (V), diversity (D), joining (J) segments and complementarity-determining region (CDR) 3 length. We found only a limited number of shared clonal families in most patients (Fig. 6b). In contrast, many of the plasma cells were clonally related, comprising 15 clonal families in patient 6, and 2 clonal families in patient 7.

Many of the clones identified in plasma cells were also present in the corresponding B cell populations. This was true for eight clones in patient 6 and one clone in patient 7. The clonal relatedness between Bin cells and plasma cells indicates that local self-antigens can drive in situ selection and differentiation into antibody secreting cells.

**Table 1 Distribution of sequenced antibody heavy chains across patients.**

|  | Kidney 1 | Kidney 2 | Kidney 3 | Kidney 4 | Kidney 5 | Kidney 6 | Kidney 7 | Tonsil 1 | Tonsil 2 | Total |
|---|---|---|---|---|---|---|---|---|---|---|
| Switched | 11 | 42 | 31 | 7 | 28 | 92 | 17 | 7 | 20 | 255 |
| Unswitched | 5 | 90 | 23 | 51 | 41 | 11 | 8 | 32 | 18 | 279 |
| Total sequenced mAb | 16 | 132 | 54 | 58 | 69 | 103 | 25 | 39 | 38 | 534 |
| Expanded clonal families | 0 | 1 | 2 | 1 | 0 | 15 | 2 | 0 | 0 | 21 |

Class-switching states and the number of expanded clonal families in sequenced antibody heavy chains are shown.

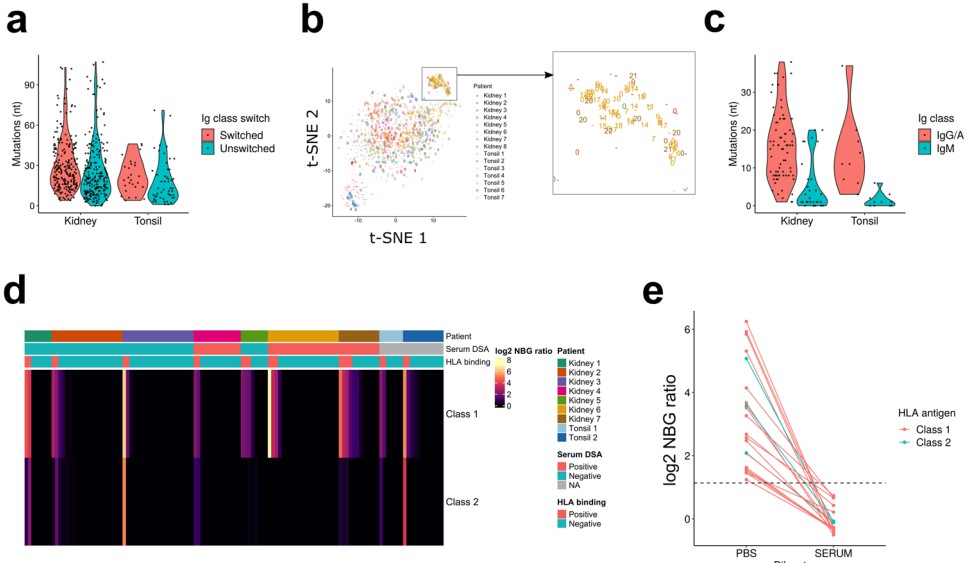

**Fig. 6 Antibodies generated by intrarenal B cells were not selected for allo-HLA reactivity. a** A violin plot showing distribution of mutations in the variable region of immunoglobulin heavy chains, grouped by tissue source and the Ig class-switch state. **b** t-SNE plots as in Fig. 6, with data points labeled with their clonal family. "-" means antibody genes were not sequenced; "0" means the cells did not share their clonotype with others; and 1-22 indicate that the cells shared a clonotype with other B cells with the same number. **c** A violin plot showing distribution of mutations in recombinantly expressed antibodies, grouped by tissue source and Ig class-switch state. **d** A heatmap showing the HLA-binding assay result. Each column represents each antibody, and maximum NBG ratio within class-1 or class-2 beads are shown. The header label indicates patients from whom the antibodies were derived, serum DSA positivity, and whether the observed reactivity was above the positivity threshold. **e** A plot showing the change in HLA binding when the antibodies were tested in negative control human serum. All the positive intrarenal antibodies in (**d**) were tested, and data points of the same antibodies were connected by lines.

In order to characterize their immunoreactivity, we next expressed the cloned immunoglobulin genes as recombinant antibodies with a FLAG tag at the heavy-chain C-terminus[51]. In total, we expressed 105 antibodies (74 IgG/A and 31 IgM) from intrarenal B cells isolated from 7 patients (Table 2, Supplementary Data 3). This included 11 out of the 21 expanded IgG plasma cell clones. We also expressed 19 antibodies from the two tonsillectomy patients (9 IgG and 10 IgM). As expected, the mutation rate was higher in IgG/IgA tonsil antibodies than IgM (Fig. 6c).

**In situ Bin cells are not selected for alloreactivity.** We first tested reactivity of in situ expressed antibodies to HLA antigens using a Luminex-based assay in which beads were coated with a mixture of HLA class-I or class-II antigens[52]. Antibody binding was evaluated by the normalized background (NBG) ratio, which is fold-increase binding over negative control serum. Binding was positive if it was equal to or higher than 2.2. When diluted in phosphate-buffered saline (PBS), 17% (18/105) bound to screening HLA beads (Fig. 6d). In serum DSA-positive patients, 9 of 47 (19%) expressed monoclonal antibodies (mAbs) were HLA-binding, while 16% (9/58) of mAbs from DSA-negative patients

bound HLA ($p = 0.82$, chi-squared test). However, we observed 21% (4/19) positivity in tonsil antibodies. Collectively, these data suggested that HLA reactivity was not a specific feature of in situ expressed antibodies from DSA-positive patients.

To further delineate the specificity of HLA binding, mAbs that bound to the screening HLA beads were tested on single HLA-antigen beads (SAB). Out of 18 tested antibodies, 17 antibodies showed trimmed mean fluorescent intensity (MFI) higher than 1,000. Unexpectedly, 15 of the 17 antibodies bound to HLA-C, with HLA-Cw*06:02 being most common (Supplementary Fig. 5a and Supplementary Data 4). Furthermore, 16 of 17 antibodies bound to multiple HLA antigens. These HLA alleles recognized by the mAbs were not donor-specific HLAs. Indeed, except for 6-2D3, all bound most strongly to recipient-expressed HLAs (Supplementary Fig. 5b).

We next examined whether shared eplets (discrete epitopes) between multiple HLA alleles could explain the unexpected broad HLA specificity[53]. However, 6-2D3 mAb had reactivity with several HLA-A and C antigens (Supplementary Fig. 5c) that did not share any common eplets. Indeed, the majority of mAbs (11 out of 17) that bound multiple HLA Class I alleles did not share an eplet among all their top-10 hits (Supplementary Fig. 5d). For

**Table 2 Isotypes of recombinantly produced antibodies.**

|       | Kidney 1 | Kidney 2 | Kidney 3 | Kidney 4 | Kidney 5 | Kidney 6 | Kidney 7 | Tonsil 1 | Tonsil 2 | Total |
|-------|----------|----------|----------|----------|----------|----------|----------|----------|----------|-------|
| IgG   | 5        | 13       | 10       | 4        | 3        | 21       | 4        | 2        | 7        | 69    |
| IgA   | 2        | 2        | 5        | 2        | 2        | 0        | 1        | 0        | 0        | 14    |
| IgM   | 1        | 6        | 6        | 8        | 3        | 0        | 7        | 5        | 5        | 41    |
| Total | 8        | 21       | 21       | 14       | 8        | 21       | 12       | 7        | 12       | 124   |

the remaining 6 mAbs, the top-10 hits did share an eplet for bound HLA-C antigens (Supplementary Fig. 5e and Supplementary Table 4)[54–56]. Thus, shared eplets could not explain the broad reactivity of most mAbs.

Another possibility is that the broad reactivity of our mAbs represents low-affinity polyreactivity. If so, adding a non-specific blocking reagent should abrogate binding[57]. Therefore, we next retested binding of the above antibodies to the screening HLA beads in the presence of negative control human serum. Strikingly, in the presence of serum, all antibodies lost their binding to HLA antigens (Fig. 6e). In toto, our results indicate that allo-reactive antibodies are not commonly selected in situ during acute renal allograft rejection.

Polyreactivity can be associated with autoreactivity[58]. Therefore, we assayed the binding of the mAbs to human epithelial type-2 (HEp-2) cells by immunofluorescence microscopy. For the mAbs generated from non-plasma Bin cells, the frequency of HEp-2-reactive clones was 18% (15/84) (Fig. 7a), which is similar to that reported for naive or tonsil GC repertoires[59,60]. HEp-2 reactivity was slightly more common in HLA cross-reactive antibodies (27%, 4/15 vs. 16%, 11/69), but this difference was not statistically significant ($p = 0.54$, chi-squared test). These results suggest these antibodies are not selected for reactivity to ubiquitous self-antigens.

In marked contrast, when we examined mAbs from clonally expanded plasma cells from patient 6, 76% of (16/21) antibodies had HEp-2 reactivity (Fig. 7a and Supplementary Fig. 6a). Reactivity was broadly distributed across eight different clonal families with many binding the nucleolus (Fig. 7b and Supplementary Fig. 6a). Those multiple clones had remarkably similar patterns on staining suggested that different in situ clonally expanded plasma cells were targeting the same or similar antigens.

**In situ selection by organ-restricted or inflammation-associated antigens**. To identify potential antigens, we selected three antinuclear antibodies (6-2A4, 6-2B9, and 6-1B5) from different clonal families (Fig. 7b): 6-A4 represented the most expanded clonal family; 6-2B9 showed the strongest signal in HEp-2 staining; and 6-1B5 showed the most specific nucleoli binding. Of the three antibodies tested, 6-2A4 and 6-2B9 showed similar broad immunoreactivity to HEp-2 nuclear antigens with relative molecular weight >50 kDa (Supplementary Fig. 6b). The 6-1B5 mAb did not detectably bind.

Next, we performed immunoprecipitations (IP) with the three mAbs from the lysates of HEp-2 nuclear fractions. Two negative control mAbs were included: 7-1A5 and 7-1E3 which did not bind to HEp-2 cells. Immunoprecipitations were resolved by sodium dodecyl sulfate–polyacrylamide gel electrophoresis (SDS-PAGE) and regions above IgG heavy chain (> 50 kDa) were excised and subjected to tandem mass spectrometry. Detected peptides were mapped to the human proteome according to the UniProt database[61]. Fold changes of signal intensity (normalized to median) were calculated for the three antinucleolar antibodies over the mean intensity of the two negative control antibodies.

For two of the three positive mAbs tested (6-2A4 and 6-2B9), the nucleolar antigen Ki-67 was the top hit (Fig. 7c). For the other antibody, 6-1B5, another nucleolar antigen HEATR1 was the top hit. Interestingly, HEATR1 was also found in the top 10 hits of 6-2B9, and thus it is possible that these antibodies were selected for the same protein complex.

We next examined if 6-2A4 and 6-2B9 directly recognized Ki-67. As Ki-67 is large (3,256 amino acids), we expressed seven recombinant fragments in total covering the whole length of Ki-67 in *E. coli*. Whole bacterial lysates were resolved by SDS-PAGE and tested for antibody binding by western blot. A commercial anti-Ki67 antibody preferentially bound fragment 3 (aa 994-1489) with less binding to other fragments (Supplementary Fig. 6 c). In contrast, both 6-2A4 and 6-2B9 bound to fragment 6 (aa 2446-2940) with 6-2A4 also binding to fragment 7 (aa 2927-3256). These data demonstrate that both in situ selected antibodies directly bound similar Ki-67 domains.

Since Ki-67 was detected by two independent clones, we searched for other anti-Ki-67 antibodies by staining tonsil tissue. We detected a total of six antibodies that showed colocalization with Ki-67, including 6-2A4 and 6-2B9 (Fig. 7d). These additional four antibodies belonged to the same clonotypes as either 6-2A4 or 6-2B9 indicating their clonal selection for Ki-67 reactivity. These results suggest that breach of self-tolerance and strong selection for self-antigens can occur in the kidney of renal allografts during rejection.

The clonal relatedness between B cells and plasma cells in patient 6 indicated that self-reactivity can occur in the local B cell pool. Furthermore, these clonally related B cell precursors expressed highly mutated antibodies suggesting that selection might occur before detectable clonal expansion. We postulated that these putative selecting antigens would be locally expressed in the inflamed kidney. Therefore, we expressed 28 representative highly mutated antibodies expressed by intrarenal B cells from patients 1–5 and 7 (4–5 antibodies per patient) (Supplementary Data 3). We then assessed if they bound inflamed allograft or normal kidney. The epitope-tagged immunoglobulin heavy chains allowed us to discern between monoclonal antibody binding and endogenous immune complexes.

Out of 28 tested antibodies, six antibodies, from five different patients, showed detectable binding to inflamed allograft renal tissue (Fig. 8). All antibodies showed specific distributions of staining with nuclear or perinuclear binding being common. However, only 4-2C3 showed nearly ubiquitous staining with the other antibodies only binding some cell types, often tubules. These observations suggest selection by specific intrarenal expressed antigens.

Four of the six antibodies (3-1E12, 4-2E2, 5-1C3, and 7-1B2) were HEp-2-reactive with three having predominately cytoplasmic staining. However, none of the HEp-2 patterns were predictive of binding to renal tissue. This was striking for 3-1E12 which bound specifically to a subset of renal cells yet gave a diffuse cytoplasmic HEp-2 pattern, and 4-2C3 which bound renal nuclei diffusely yet was HEp-2 negative. These data indicate that the HEp-2 could be misleading in evaluating tissue-specific autoantibodies. Furthermore, these data are consistent with

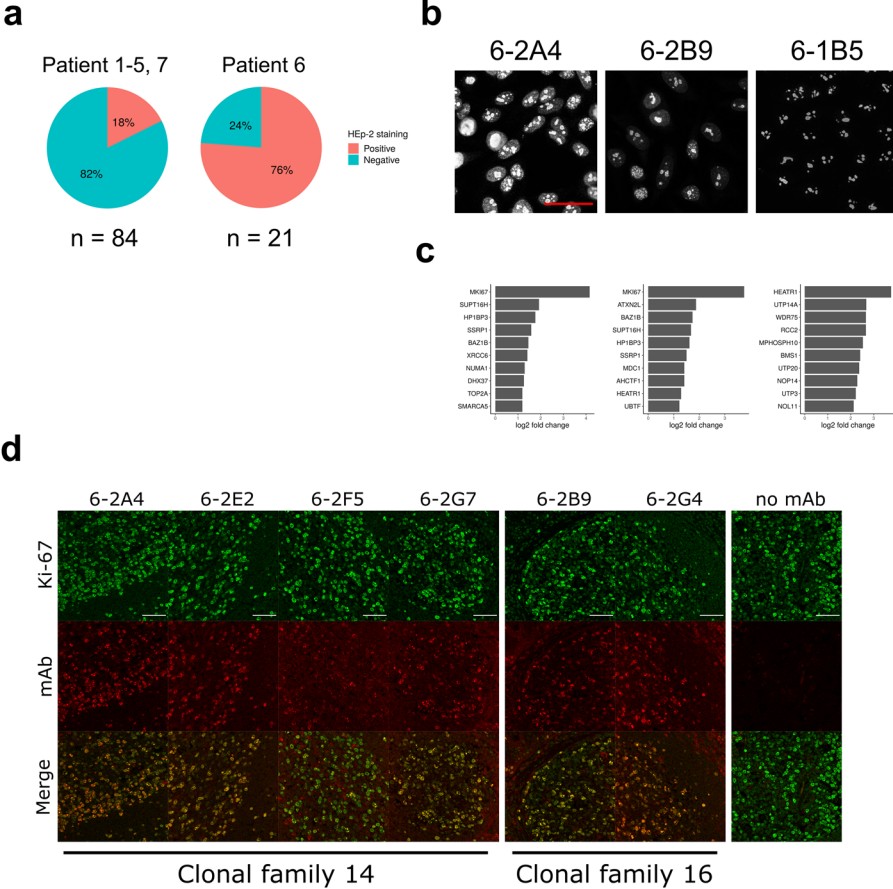

**Fig. 7 Plasma cells produce antibodies specific for nucleolar proteins including Ki-67. a** Pie charts showing the frequency of HEp-2-reactive antibodies in indicated patients. **b, c** HEp-2 staining images of the three antibodies used for the IP/mass spectrometry (**b**) and their top 10 preferentially bound antigens (**c**). Log2 fold changes of signal intensity compared with the mean of the two negative control antibodies are shown. The scale bar indicates 50 μm. **d** Staining images showing signal colocalization between the antinucleolar antibodies and a commercial anti-Ki-67 antibody on human tonsil tissue. Scale bars indicate 50 μm. For (**b, d**), staining has been repeated twice and a representative result is shown.

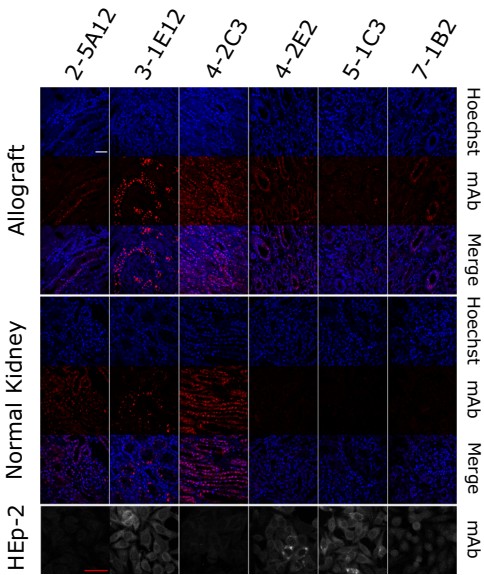

**Fig. 8 Bin cells generate IgG antibodies that bind renal antigens.** Indicated Flag-tagged antibodies were used to probe inflamed allograft or normal renal tissue. Antibodies were also assayed for HEp-2 immunoreactivity. Images are representative (*n* = 2). The white scale bar for tissue and red scale bar for HEp-2 cells both indicate 50 μm.

in situ immunity directed against renal, and non-ubiquitous, expressed antigens.

We next examined if targeted antigens were preferentially expressed in inflamed kidney. Indeed, three antibodies (4-2E2, 5-1C3, and 7-1B2) did not bind normal kidney. However, the other three antibodies demonstrated similar binding patterns in inflamed and normal renal tissue. Collectively, these data suggest a broad loss of tolerance in the intrarenal B cell compartment and selection by renal-restricted or inflammation-associated antigens.

## Discussion

Herein, we demonstrate that in allograft rejection, intrarenal B cells have a unique gene expression profile that most closely resembles the B1 innate B cells that have been described in mice but not in humans[62,63]. In contrast to murine innate B cell populations, which are constitutively present, Bin cells were associated with inflammation and overall transcriptional differences were most prominent in those cells expressing mutated IgG antibodies[62]. These data suggest that in humans, B cell innate-like features are induced by specific pathways of activation and selection.

Remarkably, within those infiltrating Bin cells expressing highly mutated class-switched antibodies, specificity for organ and inflammation-restricted antigens was frequent. This was observed across several patients who had no history of autoimmune disease. In contrast to ubiquitous self-antigens, central tolerance does not necessarily deplete the B cell repertoire of

sequestered or peripherally restricted antigen reactivity[64–69]. Furthermore, organ-restricted specificities that arise during the germinal center response are not efficiently eliminated[70]. Rather, such specificities are tolerized by mechanisms, including anergy, which allow for persistence in the periphery[71]. In mice, it is clear that innate signals and patterns are sufficient to subvert peripheral tolerance and induce systemic autoimmunity[23–26]. Herein, we demonstrate that in human allograft rejection such breaches in tolerance can occur locally at inflammatory sites rich in innate signaling networks.

One of the major differences between B cell populations in humans compared to mice is the lack of cells with an innate B1 cell transcriptional signature. While innate-like functions have been ascribed to human B cell subpopulations, extensive scRNA-seq has failed to confirm their existence[63,72]. Indeed, it was difficult to map Bin cells to known populations of human B cells. Rather, to characterize these cells we relied on transcriptional profiles of murine B1 innate cells.

Using this approach, we identified *AHNAK* and *AHNAK*-covariant genes as a defining signature of Bin cells. While its role in B cells is unclear, AHNAK has diverse functions including mediating T-cell-receptor induced intracellular calcium mobilization[73]. It also modulates TGF-β/Smad signaling, which is important for several B cell functions[74,75].

The proinflammatory cytokine IL-15 was expressed by Bin cells. A component of its receptor complex, IL-15RA, was also expressed in rejected renal allograft tissue. IL-15 can act directly or via trans-presentation by IL-15RA[76]. The abundance of IL-15 in rejected renal allografts has been long known and antagonization of IL-15 improves graft survival[77,78]. However, this is the first report to suggest that intrarenal B cells are a significant source of IL-15 in allograft rejection.

Bin cells also expressed the type-I IFN signature[79,80]. IFN-α treatment can induce rejection of renal allograft and the IFN pathway is upregulated in rejecting renal allografts[81,82]. IFN signaling induces TLR expression and indeed, *TLR2* and *TLR7* were upregulated in Bin cells. In mice, recipient expression of TLR2 and TLR4 is critical for renal allograft rejection[83]. The IFN pathway likely reflects activation mechanisms independent of the AHNAK program as there was not a correlation between the IFN and AHNAK signatures in single Bin cells. Therefore, multiple activation pathways likely contribute to the molecular state of intrarenal Bin cells.

Renal allograft rejection is associated with the presence of serum DSAs. However, despite an extensive analysis, we could not identify even one Bin cell that expressed antibodies specific for HLA, be they donor-specific or otherwise. This indicates that B cells expressing DSAs are rarely selected for in the kidney[21]. Rather, it is likely that DSAs are a manifestation of systemic alloimmunity associated with endovascular immune complex deposits[18].

Bin cells expressing organ-specific antibodies were observed in the absence of discernable clonal expansion. This suggests some aspects of selection might occur peripherally with Bin cells then homing to the inflamed organ. Alternatively, selection could be entirely local and our inability to detect clonality reflects the small sample size intrinsic to clinical samples. Regardless, our data demonstrate that observing clonal expansion is an insensitive surrogate for local antigen selection.

In the two patients in which we captured substantial plasma cell populations, there was clonal relatedness with Bin cells indicating that local antigens can drive both selection and differentiation in situ. Extensive characterization of multiple plasma cell antibodies from one patient (patient 6) revealed that they bound nucleolar antigens with two, from different clonal trees, binding directly to Ki-67. Ki-67 is associated with cell proliferation and therefore would be preferentially expressed in inflamed but not normal kidney in which the vast majority of cells are quiescent.

Serum autoantibodies are associated with allograft rejection. Many of the targeted antigens are cryptic and only exposed after ischemia-reperfusion injury[22]. These serum autoantibodies can mediate vascular injury and accelerate graft rejection[84]. However, they are present prior to transplantation and therefore are likely natural antibodies[85]. In contrast, intrarenal Bin cells expressed highly mutated IgG autoantibodies that did not bind vascular endothelium. Therefore, both natural and acquired B cell autoimmunity, targeting different antigens and compartments, are features of renal allograft rejection.

While renal reactive B cells were both present and selected in the kidney, they only constituted a minority of all infiltrating B cells. Most in situ B cells expressed unmutated or pauci-mutated IgM antibodies and therefore are unlikely to have been strongly selected by antigen. These cells could reflect non-specific trapping of cells[86]. Another possibility is that there is preferential selection for B cells expressing low-affinity anti-renal allograft antibodies. Consistent with this, Bin cells expressed CCR6 which is a marker of low-affinity GC memory B cell precursors[87].

In summary, we demonstrate that in renal allograft rejection, infiltrating B cells have a unique transcriptional state that suggests that they are driven by, and likely contribute to, specific innate signaling pathways and networks. Furthermore, our observations, and studies in mice, suggest that this innate state of activation permits the breaking of peripheral tolerance to organ-restricted antigens and molecular patterns of inflammation[23–26]. Because of their inflammatory state, and the specificity of the antibodies they express, Bin cells bridge and integrate innate and adaptive immunity to drive local inflammation.

## Methods

**Clinical sample collection.** The protocol for patient sample collection has been approved by Institutional Review Board at the University of Chicago. All the patients signed written informed consent. Kidney biopsies were performed as an additional biopsy core from consenting patients. The presence of antibody-mediated rejection was clinically confirmed for all the sequenced transplant patients. Tonsil samples were deidentified and collected from tonsillectomy cases. All the clinical samples were collected on the day of biopsy at the University of Chicago Hospital, and approved by the Internal Review Board at the University of Chicago.

**Cell sorting.** Within 5 h after collection, tissues were minced and digested with Liberase TL (Sigma-Aldrich, 5401020001) for 15 min at 37 °C. Cells were washed and stained for 30 min at 4 °C with Calcein AM (ThermoFisher Scientific, 65-0853-78) and antibodies: PE-CD19 (ThermoFisher Scientific, clone: SJ25C1, 12-0198-42), APC-CD38 (BD Biosciences, clone: HIT2, 555462), PE-Cy7-CD45 (Thermo-Fisher Scientific, clone: HI30, 25-0459-42). Stained cells were washed, and DAPI (Thermo Fisher Scientific, D1306) was added to the single-cell suspension immediately before the samples were subjected to BD FACSAria Fusion with FACSDiva 8.0.1 for sorting. Doublets were excluded by FSC-A/FSC-H gating, and CD45 + Calcein+ DAPI- CD19 + CD38 + activated B cells were single-cell sorted into 96-well plates with catching buffer (RLT lysis buffer (Qiagen, 79216) with 1% 2-mercaptoethanol (Sigma-Aldrich, 63689-25ML-F)). Sorted cells were immediately spun down and stored at -80 °C until being processed for scRNA-seq. Data were analyzed with FlowJo 10.7.1.

**scRNA-seq.** scRNA-seq was performed following Smart-seq 2 protocol[28]. mRNA was purified from sorted cell lysates using SPRI beads (Beckman Coulter, A63987), and reverse transcribed to cDNA with ERCC spike-in controls (Thermo Fisher Scientific, 4456740). cDNA was amplified for 20 cycles using KAPA HiFi HotStart ReadyMix PCR Kit (Kapa Biosystems, KK2602). Aliquots of the amplified cDNA were also used for antibody cloning later. cDNA library was generated using Nextera XT DNA Library Preparation Kit (Illumina, FC-131-1096), pooled and sequenced with Illumina sequencer. Data were analyzed with Python 3.7.1 and R 3.6.3 as detailed below.

**Read alignment, quality control, and data integration.** For mapping sequencing reads, a human transcriptome (GRCh38) was obtained from the Ensembl database. Low-complexity regions were masked from the transcriptome using RepeatMasker

4.1.0 with "-noint -norna -qq" options[88]. The masked transcriptome was used for pseudoalignment by kallisto 0.46.1[89]. For the first cohort, poor-quality cells were excluded from the analyses if they were expressing less than 3,000 genes or more than 15,000 genes. Furthermore, to exclude cells which could be non-B cells, cells were filtered out if a sum of log-count per million (cpm) of immunoglobulin heavy chain constant region genes was below 5. Batch effects were corrected by normalizing counts to ERCC using the RUVSeq 1.16.1 with "$k = 2$" option[29]. For the second cohort, read alignment and QC was done in the same manner except that 1,000 genes were used for gene count cut off. In order to normalize the difference in sequencing depths between the first and second cohorts, SCTransform in Seurat R package 3.1.1 was applied[45,46]. The normalized data were further processed by ComBat in sva R package 3.32.1 to remove batch effects between the two cohorts[47].

**t-SNE projection and cell cluster assignment for the differential gene expression analysis**. Gene expression similarity among single cells was visualized by t-SNE plots, whose coordinates were calculated by Rtsne package 0.15. Expressed Ig isotypes were identified from the scRNA-seq data by assigning the most highly expressed Ig constant region gene. Cells were categorized as "unswitched" if their isotype was IgM or IgD, and categorized as "switched" otherwise. For the first cohort, the ERCC-normalized data were scaled by log2 cpm before making t-SNE plots. For the integrated data, clusters were assigned by Seurat, and the plasma cell cluster was identified by *PRDM1* expression. Plasma cells were removed from differential expression analyses.

**Differential gene expression analysis**. Differential expression was tested on genes expressed by at least 10% of each category to be compared. For the first cohort, raw pseudocounts were rounded and subjected to edgeR 3.26.8 with unwanted variables calculated by RUVSeq in the design matrix[90]. For the integrated data, independent two-sided t-tests were applied to expression values after ComBat. For both analyses, false discovery rate (FDR) was calculated by adjusting p values for multiple testing by the Benjamin-Hochberg method. Genes with FDR ≤ 0.05 and log2 fold change ≥ 1 were categorized as differentially expressed.

**Hierarchical clustering of DEG**. For the first cohort, differential expression was tested in four comparisons (class-switched renal vs. tonsil, unswitched renal vs. tonsil, renal class-switched vs. unswitched, and tonsil class-switched vs. unswitched). Mean expression values of identified 2855 differentially expressed genes (DEGs) were calculated in four populations (renal switched, renal unswitched, tonsil switched, tonsil unswitched). Then hierarchical clustering was performed based on their expression pattern across the four population means, identifying six gene clusters. A heatmap was produced using pheatmap R package 1.0.12 based on the clustering and Z-scores calculated from the mean values.

**Pathway enrichment analysis**. GO and KEGG enrichment was tested using clusterProfiler 3.12.0 and org.Hs.eg.db annotation database 3.8.2. FDR 0.05 was used for significance cutoff[91]. When there were more than 10 significantly enriched GO terms, redundant terms were removed using "simplify" function in clusterProfiler library with its default setting.

**Enrichment analysis of AHNAK-covariant genes**. Gene expression in mouse B-cell subsets in the spleen or peritoneal cavity were fetched from Immgen[36]. Genes were identified as *Ahnak*-covariant genes, when their expression pattern within the peripheral B-cell subsets had a correlation coefficient ≥ 0.8 with *Ahnak*. The *Ahnak*-covariant genes were converted to their human orthologs as *AHNAK*-covariant genes using Ensembl database[92]. Then enrichment of the *AHNAK*-covariant genes in DEG clusters was tested by hypergeometric test. For the background frequency, we used the frequency of the *Ahnak*-covariant genes within all the mouse genes detected in Immgen microarray data.

**Correlation test of gene expression**. For Supplementary Fig. 4e, f, plasma cells were removed from the analysis. For the cells of interest, 1,000 most variant genes were identified. For Supplementary Fig 4f, we used 736 genes which were within top 1,000 high-variant genes in both the first cohort dataset and integrated dataset. Mean expression values within each cell population were calculated for the selected genes, and their Pearson correlation between populations was tested.

**Calculation of gene expression scores**. Geneset-based scores were calculated as a sum of scaled expression values of genes present in each geneset. For DEG cluster scores in mouse B-cell subsets, DEGs in each gene cluster were converted to mouse orthologs in the same manner described above. Then, mean score for the mouse genes were calculated for each replicate in Immgen data. A mean score was calculated for each B-cell subset, scaled to Z-scores and visualized as a heatmap. For innate immune genes, genes tagged to "innate immune response" GO term were identified in the DEG clusters, and used to calculate a score. For DN-associated genes, 17 DN-upregulated genes and 9 DN-downregulated genes were defined according to Arazi et al., 2019[37]. Then the difference between a scaled sum of expression of DN-upregulated and downregulated genes was used as the score.

**Tissue staining**. Paraffin-embedded formalin-fixed tissue blocks were sectioned by 3 μm thickness. Tissue sections were deparaffinized with xylene and ethanol, and subjected to antigen retrieval with 10 mM citrate buffer pH 6.0 (ThermoFisher Scientific, 005000). Tissue sections were blocked with Tris-buffered saline (TBS) containing 10% normal donkey serum (Jackson ImmunoResearch Laboratories, 017-000-121), and incubated with a combination of primary antibodies: rat or rabbit anti-CD19 (Invitrogen, clone: 6OMP31, 14-0194-82, or abcam, clone: EPR5906, ab134114, respectively, 1:100), rabbit anti-AHNAK (Proteintech, 16637-1-AP, 1:100), mouse anti-IL15 (abcam, ab55276, 1:50), and rabbit anti-Ki-67 (abcam, clone: EPR3610, ab92742, 1:100). Antibody binding was detected by 1:1000-diluted fluorophore-conjugated highly cross-adsorbed secondary antibodies from ThermoFisher Scientific (Alexa Fluor 488 donkey anti-rat IgG, A21208; Alexa Fluor 488 donkey anti-rabbit IgG, A21206; Alexa Fluor 594 donkey anti-mouse IgG, A21203; Alexa Fluor 647 donkey anti-rabbit IgG, A31573; Alexa Fluor 647 Plus donkey anti-goat IgG, A32849), and nuclei were stained with Hoechst 33342 (ThermoFisher Scientific, H3570, 1:500). For staining of FLAG-tagged recombinant antibodies cloned from rejection patients, rat anti-FLAG (BioLegend, clone: L5, 637301, 1:200) was used as the secondary antibody, which was then detected with fluorophore-conjugated anti-rat IgG antibodies. Stained sections were mounted in ProLong Gold Antifade Mountant (ThermoFisher Scientific, P36934) and imaged on SP8 confocal microscopy (Leica). Data were analyzed by ImageJ 2.0.0.

**Antibody cloning and recombinant expression**. Variable regions of antibody heavy and light chain genes were amplified from cDNA using nested PCR[48]. PCR products were Sanger-sequenced and mapped to IMGT reference. Results of heavy-chain genes were analyzed for clonality and mutation frequency. Clonal families were defined by VDJ gene usage and CDR3 length. Next, heavy and light chain variable regions were cloned into an IgG expression vector. The vector had a FLAG tag at the C terminus of IgG constant region to enable tissue staining in the presence of IgG-expressing cells or IgG deposition. A pair of cloned heavy and light chain vectors were transfected to HEK293 cells, and expressed IgG was purified using Protein A agarose beads (ThermoFisher Scientific, 20334), eluted in 0.1 M glycine-HCl pH 2.8, neutralized with 1 M Tris buffer pH 9.0 and stored in PBS with 0.05% sodium azide.

**HLA-binding assay**. Antibodies were diluted at 150 μg/mL in PBS and tested on LAB Screen Mixed (OneLambda, LSM12) according to the manufacturer's protocol. To test the binding in the presence of blocking, positive antibodies from intrarenal B cells were retested in the presence of human serum proteins. Antibodies were prepared in PBS, then diluted 1:1 in PBS or negative control serum included in the kit, and subjected to the assay. NBG ratio was calculated as the experimental readout according to the manufacturer's protocol:

$$\text{NBG ratio}_i = \frac{S_i - S_0}{N_i - N_0} \quad (1)$$

$S_i$ and $S_0$ are sample signals from the *i*th antigen-coated beads and negative beads, and $N_i$ and $N_0$ are negative serum signals from the *i*th antigen-coated beads and negative beads. NBG ratio ≥ 2.2 was used as the positivity threshold. To plot log2-transformed values, NBG ratio less than 1 was replaced with 1. For SAB assay, differences in trimmed MFI between antibodies and negative control serum were used as a readout. Eplets information was fetched from HLA Epitope Registry[93],

**HEp-2 cell staining**. Antibodies were diluted at 50 μg/mL in PBS, and tested on NOVA Lite HEp-2 ANA kit (Inova Diagnostics, 708100) according to the manufacturer's protocol. Antibody binding was detected on SP8 confocal microscopy by fluorescent signal from fluorescein isothiocyanate (FITC)-conjugated polyclonal anti-human IgG secondary antibody included in the kit.

**Western blot**. HEp-2 cells (ATCC, CCL-23) were cultured and harvested. A nuclear fraction was prepared using Nuclear Extraction Kit (Abcam, ab113474) following the manufacturer's protocol. Before the final centrifugation step, lysates were sonicated with three times of a 10-second pulse on ice. Lysates were boiled in Laemmli buffer at 95 °C for 5 min, and resolved by SDS-PAGE. Proteins on the gel were transferred to a polyvinylidene fluoride membrane, blocked by 5 % bovine serum albumin-containing TBS, and incubated with 10 μg/mL antibody diluted in the blocking buffer at 4 °C overnight. The membrane was washed and incubated with a horseradish peroxidase-conjugated anti-human IgG antibody (ThermoFisher Scientific, 31413, 1:10,000), and binding was detected with Pierce ECL Western Blotting Substrate (ThermoFisher Scientific, 32209).

**Expression of Ki-67 fragments**. Seven fragments were designed to cover the whole Ki-67 protein. Fragments 1 and 2 were cloned by PCR from cDNA of HEp-2 cells. DNA encoding the other fragments were purchased from Bio Basic Inc. Primers used and DNA fragments ordered are listed in Supplementary Data 5. DNA encoding each fragment were digested with NdeI and XhoI restriction enzymes (NEB, R0111S, and R0146S) and cloned into pET-24b (+) vector. Rosetta (DE3) E. coli (Millipore, 70954-3) were transformed with the expression vectors. Overnight culture was diluted in 1:5 in LB media, and IPTG was inoculated at 0.5 mM. After 4 h of culture, 1 mL of culture was

centrifuged, and cell pellet was resuspended in RIPA lysis buffer. The lysate was used for western blot.

**IP from a nuclear fraction of HEp-2 cell lysates.** HEp-2 nuclear lysates were precleared with Protein A Agarose beads, and incubated with 5 µg of antibodies at 4 °C overnight. The beads were washed with Tween20-containing PBS, and captured antibody-antigen complexes were eluted and resolved by SDS-PAGE. Gels were stained with InstantBlue Protein Stain (Abcam, ab119211) at 4 °C overnight. Stained gels were destained in deionized water, excised leaving molecular weight higher than IgG heavy chain, and used for mass spectrometry.

**Sample preparation for mass spectrometry.** Gel Samples were excised by sterile razor blade and chopped into ~1 mm$^3$ pieces. Each section was washed in distilled water and destained using 100 mM $NH_4HCO_3$ pH 7.5 in 50% HPLC-grade acetonitrile (VWR, BJ015-4). Samples were reduced by adding 100 µL 50 mM $NH_4HCO_3$ pH 7.5 and 10 µL of 200 mM tris (2-carboxyethyl) phosphine HCl and incubating at 37 °C for 30 min. The proteins were alkylated by adding 100 µL of 50 mM iodoacetamide freshly prepared in 50 mM $NH_4HCO_3$ pH 7.5 buffer and incubated in the dark at 20 °C for 30 min. Gel sections were washed in water and acetonitrile, and vacuum dried. Samples were digested by sequencing-grade modified trypsin (Promega, V5111) in 50 mM $NH_4HCO_3$ pH 7.5, and 20 mM $CaCl_2$ at with 1:50–1:100 enzyme–protein ratio for overnight at 37 °C. Peptides were extracted first with 5 % formic acid, then with 75 % ACN:5% formic acid, combined and vacuum dried. Digested peptides were cleaned up on a C18 column (PicoFrit column blanks (New Objective, PF360-100-15-N-5) self-packed with Agilent Poroshell 120, EC-C18, 2.7 µm (Agilent 696975-902)), speed vacuumed and sent for liquid chromatography–tandem mass spectrometry (LC-MS/MS) to the Proteomics Core at Mayo Clinic.

**High-performance liquid chromatography for mass spectrometry.** All samples were resuspended in Honeywell Burdick & Jackson HPLC-grade water (VWR, BJ365-4) containing 0.2 % formic acid (ThermoFisher Scientific, PI28905), 0.1 % TFA (ThermoFisher Scientific, PI28904), and 0.002 % Zwittergent 3-16 (Calbiochem, 603023). This sulfobetaine detergent contributes to the following distinct peaks at the end of chromatograms: $MH^+$ at 392, and in-source dimer $[2 M + H^+]$ at 783, and some minor impurities of Zwittergent 3-12 seen as $MH^+$ at 336. Peptide samples were loaded to a 100 µm × 40 cm PicoFrit column self-packed with 2.7 µm Agilent Poroshell 120, EC-C18, washed, then switched in-line with a 0.33 µL Optimize EXP2 Stem Traps spray tip nano column packed with Halo 2.7 µm Pep ES-C18 (Optimize Technologies, 15-04001-HN) for a 2-step gradient. Mobile phase A was water/acetonitrile/formic acid (98/2/0.2) and mobile phase B was acetonitrile/isopropanol/water/formic acid (80/10/10/0.2). Using a flow rate of 350 nL/min, a 90-min 2-step LC gradient was run from 5% B to 50 % B in 60 min, followed by 50–95% B over the next 10 min, hold 10 min at 95 % B, back to starting conditions and re-equilibrated.

**LC-MS/MS data acquisition and analysis.** The samples were analyzed by data-dependent electrospray tandem mass spectrometry (LC-MS/MS) on a Thermo Q-Exactive Orbitrap mass spectrometer, using a 70,000 RP survey scan in profile mode, $m/z$ 360–2,000 Da, with lockmasses, followed by 20 HCD fragmentation scans at 17,500 resolution on doubly and triply charged precursors. Single charged ions were excluded, and ions selected for MS/MS were placed on an exclusion list for 60 s.

All LC-MS/MS *.raw Data files were analyzed with MaxQuant 1.5.2.8, searching against the UniProt Human database (Download on 9/16/2019 with isoforms, 192,928 entries) *.fasta sequence, using the following criteria: LFQ was selected for Quantitation with a minimum of 1 high confidence peptide to assign LFQ Intensities. Trypsin was selected as the protease with maximum missing cleavage set to 2. Carbamidomethyl (C) was selected as a fixed modification. Variable modifications were set to Oxidization (M), Formylation (N-term), Deamidation (NQ). Orbitrap mass spectrometer was selected using an MS error of 20 ppm and a MS/MS error of 0.5 Da. 1 % FDR cutoff was selected for peptide, protein, and site identifications. Ratios were reported based on the LFQ Intensities of protein peak areas identified by MaxQuant and reported in the proteinGroups.txt. The proteingroups.txt file was processed in Perseus 1.6.7. Proteins were removed from this results file if they were flagged by MaxQuant as "Contaminants", "Reverse" or "Only identified by site". Three biological replicates were performed. Samples were filtered to require hits to have been seen in at least two replicates per condition. Intensities were normalized by median intensity within each sample. Then, log2 fold changes over the means of negative controls were obtained for the three antinucleolar antibodies.

**Reporting summary.** Further information on research design is available in the Nature Research Reporting Summary linked to this article.

## Data availability

RNA sequencing and mass-spectrometry data have been deposited in the GEO database under primary accession code (GSE162809) and PRIDE Archive under primary accession code (PXD022989), respectively. Source data are provided with this paper.

## Code availability

All the code written to analyze and visualize data using publicly available programs is available at https://github.com/yutasano0121/Asano_NC_2021.

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

## Acknowledgements

This work was supported by the NIH Autoimmunity Centers of Excellence and NIH grant U19 AI082724 (M.R.C. and S.P.) and NIH grant AI148705 (M.R.C. and A.C.). The authors acknowledge Mayo Clinic Medical Genome Facility Proteomics Core for generating raw mass-spectrometry data. Y.A. also acknowledges Heiwa Nakajima Foundation for its support.

## Author contributions

Y.A. and M.C. were involved in the conception and study design. Y.A., M.J., and P.C. collected patient samples. AChang and K.K. handled clinical information. Y.A. isolated single B cells. J.D. generated scRNA-seq data with SP's supervision. Y.A. and A.A.K. analyzed transcriptomic data. Y.A. and A.K. generated antibodies. Y.A. and M.V. performed staining and imaging analysis. D.J. tested HLA binding of recombinant antibodies. Y.A. and A.T. analyzed eplet mapping. Y.A. and D.W. generated and analyzed mass-spectrometry data. Y.A., AChong, and M.C. interpreted data and results of analysis. Y.A. and M.C. prepared the manuscript. Critical reading and intellectual assessment of the manuscript were done by all the authors. All authors read and approved the final manuscript. All authors contributed to the article and approved the submitted version.

## Competing interests

The authors declare no competing interests.
