## [Peer Review File · Nature Communications]

REVIEWER COMMENTS

Reviewer #1 (Remarks to the Author):

In this study, dr. Asano and colleagues studied the contribution of intrarenal B cells to allograft rejection. They used scRNA-seq from sorted activated B cells in late renal allograft biopsies and human tonsils as controls. Comparison across tissue sources (kidney, tonsil) and IgG class switch status (switched, unswitched) revealed 2855 differentially expressed genes which formed 6 clusters. Two of them (cluster 2 and 3) represented genes enriched in grafts and were related mainly to innate immune response and IFN-related pathways. Of interest they noted clearly higher expression of AHNAK mRNA from cluster 3 in grafts compared to tonsils. Cluster 3 resembles a gene signature of peritoneal B1 cells (Bin) previously observed in mice but not yet in human. Expression profiles of activated B cells did not differ between DSA+ve (n=3) and DSA-ve biopsies (n=5). Moreover authors evaluated intragraft B cells in regards of alloreactivity. They cloned immunoglobulin genes as recombinant antibodies and tested their HLA binding in situ by Luminex and suggested that alloreactive antibodies are not commonly selected C4d/DSA+ grafts. Interestingly, authors found that Bin cells expressed antibodies reactive with several renal and inflammation-associated antigens. Such local antigens may drive Bin cell proliferation and differentiation into plasma cells and thus contribute to tissue injury. Therefore local autoimmune mechanisms are likely to contribute, besides DSA driven alloimmunity, to graft injury during chronic ABMR.

The manuscript is well written, technically sound and has several priorities. Authors first identified Bin cells in human and described its possible pathophysiological role during allograft pathology. The contribution of local autoimmunity to allograft injury remains however unknown as the studied population is extremely small (which is understandable in such kind of research) and yet poorly defined.

There are some questions and comments:

Biopsies were performed late, some biopsies were C4d+/DSA+, others not. Please add Banff grades in all biopsies along with the information about graft function (Cr, eGFR, proteinuria) and follow-up, I guess that you mean by graft survival the transplant follow-up, please conform. Is it possible that described autoimmune mechanisms are associated with advanced IFTA rather than alloimmunity (when biopsies were performed late..)?

Are Bin cells kidney specific or might be found in other grafted organs where B cells infiltration might be found?

When B cell driven autoimmunity contributes to graft injury independently on known alloimmune mechanisms, could such mechanisms be found in not-grafted organs during other pathologies? In the entire text you write about kidney allograft rejection but you used late biopsies with unclear rejection status, I guess half of biopsies were chronic (active?) ABMR and second half was IFTA. Therefore also the title of the manuscript needs to be adapted accordingly.

Reviewer #2 (Remarks to the Author):

Clark and colleagues evaluate the single cell transcriptomes and BCR repertoire of intrarenal B cells from rejection allografts, compared to tonsillectomy. They find that intrarenal B cells, in particular switched B cells, have a distinct gene signature compared to B cells from tonsil, marked by high expression of AHNAK and a gene signature that is highly expressed in PC B1 B cells in mice. mRNA and protein expression of AHNAK is demonstrated in intrarenal B cells. Repertoire analysis identifies some highly mutated BCRs from renal B cells, including in plasma cells and in some non-plasma cells, and some repertoire overlap is identified between plasma cells and other B cells in kidney, suggesting potential in situ plasma cell differentiation. The authors express 105 BCRs from intrarenal B cells, which appeared not enriched for binding donor MHC. The authors identify ki67 through mass spec as a target of some antibody clones, and further analysis of 28 highly mutated antibodies from intrarenal B cells found 6 antibodies that binding kidney, including 3 that preferentially bind inflamed kidney over control kidney.

The data are clearly presented and provide a valuable, unique, and stimulating view into intrarenal B cells in allograft rejection. The scope – identifying B cell phenotypes and receptors, and then identifying antigen target(s), is impressive and commendable. A few questions:

1) AHNAK-associated gene signatures:

a. If one calculates an expression score for AHNAK co-variant genes (excluding AHNAK), is this set of genes upregulated in renal B cells compared to tonsil cells?

b. If one calculates a gene score for a peritoneal cavity B cell signature using the Immgen data, is expression of this gene set higher in renal B cells than tonsil B cells?

2) Connection to B cell phenotypes in other diseases

a. The authors reference scRNAseq data on B cells from lupus nephritis kidneys. An effort to overlay or connect these two analyses would be helpful. Can the intrarenal B cells described here in allograft rejection be matched to populations in Arazi et al? The authors note that it is difficult to map Bin cells to known human b cell populations – would these cells cluster uniquely compared to B cell clusters from lupus nephritis or other renal diseases if other datasets are available?

3) Antibodies. The identification of Ki67 as an antibody target is interesting, and the suggestion that there are other inflammation-associated intrarenal targets is intriguing. Are the anti-Ki67 antibodies detectable in serum? Do the 6 anti-kidney inflammation-associated antibodies described bind other tissues (inflamed or otherwise)?

MINOR: Figure S7 is conceptually important in the paper – would consider moving this to a main figure if feasible.

Reviewer #3 (Remarks to the Author):

I will focus my critique points on the data analyses, as I am not an expert in graft immunology. From my perspective, many of the analyses need to be performed more rigorously in order to provide satisfactory confidence in assertions made throughout the text.

One recurring issue arises from statements made about a number of the tSNE plots (e.g., Figure 2, Figure 5, and multiple supplementary figures), declaring that samples or cell types are or are not distributed similarly between perceived clusters; these declarations do not appear to have derived from statistical comparisons against null hypotheses. This concern is underscored, of course, by the fact that the dimensionality reduction visualization techniques nonlinearly compress potential differences from other dimensions of the data so pure observation may be misleading.

An analogous shortcoming applies to rendering conclusions from derived quantities such as in Figure 3G.

A further issue, along resonant lines though confined to Figure 4, is that there is no rigorous analysis of the ligand/receptor inferences. Mere visualization, such as here in Cytoscape, does not in any way substantiate any such ligand/receptor pair as statistically significant. Many principled algorithms have been demonstrated in the literature for determining degrees of confidence in these kinds of inferences, yet none appears to have been employed here.

Reviewer 1

“In this study, dr. Asano and colleagues studied the contribution of infrarenal B cells to allograft rejection. They used scRNA-seq from sorted activated B cells in late renal allograft biopsies and human tonsils as controls. Comparison across tissue sources (kidney, tonsil) and IgG class switch status (switched, unswitched) revealed 2855 differentially expressed genes which formed 6 clusters. Two of them (cluster 2 and 3) represented genes enriched in grafts and were related mainly to innate immune response and IFN-related pathways. Of interest they noted clearly higher expression of AHNAK mRNA from cluster 3 in grafts compared to tonsils. Cluster 3 resembles a gene signature of peritoneal B1 cells (Bin) previously observed in mice but not yet in human. Expression profiles of activated B cells did not differ between DSA+ve (n=3) and DSA-ve biopsies (n=5). Moreover authors evaluated intragraft B cells in regards of alloreactivity. They cloned immunoglobulin genes as recombinant antibodies and tested their HLA binding in situ by Luminex and suggested that alloreactive antibodies are not commonly selected C4d/DSA+ grafts. Interestingly, authors found that Bin cells expressed antibodies reactive with several renal and inflammation-associated antigens. Such local antigens may drive Bin cell proliferation and differentiation into plasma cells and thus contribute to tissue injury. Therefore local autoimmune mechanisms are likely to contribute, besides DSA driven alloimmunity, to graft injury during chronic ABMR.

The manuscript is well written, technically sound and has several priorities. Authors first identified Bin cells in human and described its possible pathophysiological role during allograft pathology. The contribution of local autoimmunity to allograft injury remains however unknown as the studied population is extremely small (which is understandable in such kind of research) and yet poorly defined.

There are some questions and comments.”

1. “Biopsies were performed late, some biopsies were C4d+/DSA+, others not. Please add Banff grades in all biopsies along with the information about graft function (Cr, eGFR, proteinuria) and follow-up.”

A: To Table S1 we have now added Banff grades along with available renal function near time of biopsy and at latest follow-up. As can be seen, there was a range of chronicity with only one biopsy sample (patient #3) having severe (3) interstitial fibrosis and tubular atrophy. Of the eight patients, only one subsequently rapidly progressed to renal graft failure.

2. “I guess that you mean by graft survival the transplant follow-up, please conform.”

A: Yes, the “graft survival” in Table S1 refers to duration since transplantation to time of biopsy. In addition, we have added additional follow-up, as available, after biopsy.

3. “Is it possible that described autoimmune mechanisms are associated with advanced IFTA rather than alloimmunity (when biopsies were performed late..)?”

A: As noted above, the biopsy samples we used had a range of IFTA with only one having severe IFTA and three having moderate IFTA. Of note, plasma cells and evidence of *in situ* B cell differentiation were observed in two patients, one with moderate and one with mild IFTA. Therefore, it is unlikely that the observed immune mechanisms are secondary to IFTA.

4. “Are Bin cells kidney specific or might be found in other grafted organs where B cells infiltration might be found?”

A: This is certainly an interesting question. However, to answer this question would require duplicating large parts of our study in a new transplant setting. Such studies are well beyond the scope of the present investigation.

5. “When B cell driven autoimmunity contributes to graft injury independently of known alloimmune mechanisms, could such mechanisms be found in not-grafted organs during other pathologies?”

A: Yes, *in situ* breaking of B cell tolerance to molecular patterns of inflammation might be a general feature of both autoimmune and non-autoimmune diseases. Indeed, in lupus nephritis, we have observed a breaking of B cell tolerance to local inflammation-associated antigens in the kidney (*Front Immunol* 11:593177, *Lupus* 29:569, *Arth Rheum* 66:3359).

6. “In the entire text you write about kidney allograft rejection but you used late biopsies with unclear rejection status, I guess half of biopsies were chronic (active?) ABMR and second half was IFTA. Therefore also the title of the manuscript needs to be adapted accordingly.”

A: As discussed above, our observations span a wide range of IFTA grades. Furthermore, five of the biopsies had evidence of active ABMR (AMR) (Table S1). As our manuscript title does not specify either acute or chronic rejection, we feel that it is appropriate.

Reviewer 2

“Clark and colleagues evaluate the single cell transcriptomes and BCR repertoire of intrarenal B cells from rejection allografts, compared to tonsillectomy. They find that intrarenal B cells, in particular switched B cells, have a distinct gene signature compared to B cells from tonsil, marked by high expression of AHNAK and a gene signature that is highly expressed in PC B1 B cells in mice. mRNA and protein expression of AHNAK is demonstrated in intrarenal B cells. Repertoire analysis identifies some highly mutated BCRs from renal B cells, including in plasma cells and in some non-plasma cells, and some repertoire overlap is identified between plasma cells and other B cells in kidney, suggesting potential *in situ* plasma cell differentiation. The authors express 105 BCRs from intrarenal B cells, which appeared not enriched for binding donor MHC. The authors identify ki67 through mass spec as a target of some antibody clones,

and further analysis of 28 highly mutated antibodies from intrarenal b cells found 6 antibodies that binding kidney, including 3 that preferentially bind inflamed kidney over control kidney.

The data are clearly presented and provide a valuable, unique, and stimulating view into intrarenal B cells in allograft rejection. The scope – identifying B cell phenotypes and receptors, and then identifying antigen target(s), is impressive and commendable. A few questions:”

1. “AHNAK-associated gene signatures. If one calculates an expression score for AHNAK co-variant genes (excluding AHNAK), is this set of genes upregulated in renal B cells compared to tonsil cells?”

A: Yes, even without *AHNAK*, there is clear and strong upregulation of *AHNAK* co-variant genes differentially expressed in intrarenal compared to tonsil B cells. This is especially true when comparing isotype switched B cells. These new data are provided in Figure S2G.

2. “If one calculates a gene score for a peritoneal cavity B cell signature using the Immgen data, is expression of this gene set higher in renal B cells than tonsil B cells?”

A: To address this question, we initially did two comparisons, peritoneal B1a vs. splenic B1a cells and peritoneal follicular vs. splenic follicular B cells. This analysis revealed 19 genes specifically upregulated in peritoneal but not splenic B cells. There was a trend towards enrichment of this gene signature in human renal but not tonsil B cells (data not shown). However, it did not reach statistical significance. As this specific comparison was not revealing, we did not include it in our revised manuscript.

3. “Connection to B cell phenotypes in other diseases. The authors reference scRNA-seq data on B cells from lupus nephritis kidneys. An effort to overlay or connect these two analyses would be helpful. Can the intrarenal B cells described here in allograft rejection be matched to populations in Arazi et al?”

A: Indeed, we did relate our results to those of Arazi and colleagues. They observed an enrichment in ABC/DN B cells in lupus nephritis. In contrast, we did not see an enrichment of this population suggesting that the B cells infiltrating the kidney in lupus and allograft rejection are different (Figure S3A). We did observe an upregulation of *AHNAK* in their intrarenal B0 cells. Therefore, there might be a limited upregulation of innate B cell genes in some lupus intrarenal B cells.

4. “The authors note that it is difficult to map Bin cells to known human b cell populations – would these cells cluster uniquely compared to B cell clusters from lupus nephritis or other renal diseases if other datasets are available?”

A: It is very difficult to combine disparate scRNA-seq data sets in a way that allows them to be compared by conventional clustering approaches. Technical differences often overwhelm any underlying biological differences. Indeed, it would be interesting to

directly compare intrarenal B cell phenotypes from different diseases in a single experiment.

5. “Antibodies. The identification of Ki67 as an antibody target is interesting, and the suggestion that there are other inflammation-associated intrarenal targets is intriguing. Are the anti-Ki67 antibodies detectable in serum?”

A: This is certainly an interesting question. However, our study was focused exclusively on dissecting intrarenal immunity. We did not obtain matched serum samples.

6. “Do the 6 anti-kidney antibodies inflammation-associated antibodies described bind other tissues (inflamed or otherwise)?”

A: In our studies, HEp-2 binding did not predict renal binding and therefore it is unlikely that these antibodies recognize ubiquitous self-antigens. To determine if they bind other tissue specific antigens, or bind to epitopes conserved across different inflammatory states, would require extensive studies that are clearly beyond the scope of this already large and exhaustive study.

7. “MINOR: Figure S7 is conceptually important in the paper – would consider moving this to a main figure if feasible.

A: Figure S7 is now Figure 8.

Reviewer 3

“I will focus my critique points on the data analyses, as I am not an expert in graft immunology. From my perspective, many of the analyses need to be performed more rigorously in order to provide satisfactory confidence in assertions made throughout the text.”

1. “One recurring issue arises from statements made about a number of the t-SNE plots (e.g., Figure 2, Figure 5, and multiple supplementary figures), declaring that samples or cell types are or are not distributed similarly between perceived clusters; these declarations do not appear to have derived from statistical comparisons against null hypotheses. This concern is underscored, of course, by the fact that the dimensionality reduction visualization techniques nonlinearly compress potential differences from other dimensions of the data so pure observation may be misleading.”

A: For Figure 2, we did mention that Ig-unswitched renal and tonsil cells clustered together on a t-SNE plot (Figure 2A, B). However, we then quantitatively analyzed differential gene expression across tissue sources and Ig class-switching states and identified differences between these two populations. We visualized these genes in a heatmap and explored their differential expression in the rest of Figure 2 (Figure 2C-J). For all these subsequent analyses, we used rigorous and appropriate statistical tests.

To further address the concerns of the reviewer, we have now provided an analysis of correlation across the different cell populations. These new comparisons, which are described below, are provided in a revised Figures S2A and S4E and F.

First, for the top 1,000 genes with highest variance in the first-cohort dataset (Figure 2), we calculated mean expression values in each cell population (kidney or tonsil, and Ig-switched or unswitched), and then tested their correlation between cell populations. Pearson's r values are shown in Figure S2A. Kidney unswitched B cells were relatively similar to kidney switched and tonsil unswitched B cells and very different than tonsil switched B cells. These differences correlated with differences and similarities seen in the t-SNE plot.

We repeated this analysis for the datasets described in Figure 5 (Figure S4E). For this analysis, we removed plasma cells. As can be seen, the same general relationships observed for the first dataset (Figure 2) held for the combined datasets.

Finally, we compared correlations between the first data set (Figure 2) and the integrated data set (Figure 5). For this analysis, correlation was tested for 736 genes which were within top 1,000 high-variance genes in both datasets (Figure S4F). Corresponding populations in the two datasets had the highest correlations. Furthermore, relative differences observed in the previous two analyses were evident here. For example, there was a relatively high correlation between kidney and tonsil unswitched B cells while the corresponding switched B cells were very different.

2. "An analogous shortcoming applies to rendering conclusions from derived quantities such as in Figure 3G."

A: We now provide statistical tests comparing cluster 3 enrichment in B1a.PC and B1b.PC cells to the other cell populations (Table S4). As can be seen, B1a and B1b had significantly higher cluster 3 gene scores than GC B cells which are the appropriate comparator population for antigen stimulated cells.

3. "A further issue, along resonant lines though confined to Figure 4, is that there is no rigorous analysis of the ligand/receptor inferences. Mere visualization, such as here in Cytoscape, does not in any way substantiate any such ligand/receptor pair as statistically significant. Many principled algorithms have been demonstrated in the literature for determining degrees of confidence in these kinds of inferences, yet none appears to have been employed here."

A: In Figure 4, we examined "co-upregulated" ligand-receptor pairs between our single cell B cell data and public tissue-level data. For the genes upregulated by intrarenal class-switched B cells, we identified those that were differentially expressed compared to class-switched tonsil cells. For the genes upregulated in rejected kidney tissue, we identified differentially upregulated genes (selected from genes encoding 1,340 differential transcripts reported in the original paper) in rejected renal tissue compared to normal renal tissue (Sarwal *et al.*). Therefore, the identification of receptors and ligands in each dataset was based on statistical tests.

We assigned receptor-ligand pairs according to FANTOM5 database. The pairings were supported by literature and independent databases such as STRING (<https://string-db.org/>) which have cataloged experimentally determined protein-protein interactions. In sum, our identification of putative receptors and ligands is implicated by co-upregulation of genes in each dataset and based on experimentally observed interactions.

We are aware of tools to infer ligand-receptor interaction in scRNA-seq data (reviewed in *Nat Genetics* 22:71). However, we identified ligand-receptor pairs in two distinct datasets (scRNA-seq of B cells and whole-tissue microarray data). Therefore, such tools are not applicable to our analysis.

REVIEWER COMMENTS

Reviewer #1 (Remarks to the Author):

Authors well responded to my questions/comments and I congratulate them for such important study which improves our knowledge in the field of kidney transplantation.

Reviewer #2 (Remarks to the Author):

I appreciate the responses and am satisfied with the manuscript.

Reviewer #3 (Remarks to the Author):

The authors have largely provided satisfactory modification of the manuscript in response to my previous critique points. The one exception is the ligand-receptor interaction section, which remains quite speculative. If I am reading the text correctly the focus here is on ligand-receptor pairs are both upregulated together, which is of course not necessary for increase of signaling through that nexus; either being upregulated could not only be sufficient but in fact more important than some interactions for which both are upregulated. Addressing this uncertainty requires some perspective on how the magnitude of the paired interaction changes, which is what much of the literature on ligand-receptor inferences tries to deal with. The experimental limitations involved in this current study, which the authors noted in their response letter, does make it difficult to do much better than what they show here. So, the question arises then as to what is gained from the speculative findings shown in Figure 4. It seems that the authors do not themselves put too much stock in these particular results since they are not included in the Discussion section.

“Reviewer #3 (Remarks to the Author)

The authors have largely provided satisfactory modification of the manuscript in response to my previous critique points. The one exception is the ligand-receptor interaction section, which remains quite speculative. If I am reading the text correctly the focus here is on ligand-receptor pairs are both upregulated together, which is of course not necessary for increase of signaling through that nexus; either being upregulated could not only be sufficient but in fact more important than some interactions for which both are upregulated. Addressing this uncertainty requires some perspective on how the magnitude of the paired interaction changes, which is what much of the literature on ligand-receptor inferences tries to deal with. The experimental limitations involved in this current study, which the authors noted in their response letter, does make it difficult to do much better than what they show here. So, the question arises then as to what is gained from the speculative findings shown in Figure 4. It seems that the authors do not themselves put too much stock in these particular results since they are not included in the Discussion section.”

A: In response to the above, and suggestions from the Editor, this analysis has been removed from the current manuscript. Instead, those molecules that were part of this analysis, and which were preferentially expressed by intrarenal B cells, are provided as violin plots (Fig. 4C).